



# Sun-induced Fluorescence and Near Infrared Reflectance of vegetation track the seasonal dynamics of gross primary production over Africa

Anteneh Getachew Mengistu[1,4], Gizaw Mengistu Tsidu[1,2], Gerbrand Koren[3], Maurits L. Kooreman[4], K. Folkert Boersma[3,4], Torbern Tagesson[6,7], Jonas Ardö[6], Yann Nouvellon[8,9], and Wouter Peters[3,5]

[1]Department of Physics, Addis Ababa University, Addis Ababa, Ethiopia
[2]Department of Earth and Environment, Botswana International University of Science and Technology, Palapye, Botswana
[3]Wageningen University, Meteorology and Air Quality Group, Wageningen, The Netherlands
[4]Royal Netherlands Meteorological Institute, De Bilt, The Netherlands
[5]University of Groningen, Centre for Isotope Research, Groningen, The Netherlands
[6]Department of Physical Geography and Ecosystem Science, Lund University, Sweden
[7]Department of Geosciences and Natural Resource Management, University of Copenhagen, Denmark
[8] Eco&Sols, Univ Montpellier, CIRAD, INRA, IRD, Montpellier SupAgro, 34060 Montpellier, France
[9]CIRAD, UMR Eco&Sols, 34060 Montpellier, France

**Correspondence:** Anteneh (antenehgetachew7@gmail.com)

**Abstract.**

The carbon cycle of tropical terrestrial vegetation plays a vital role in the storage and exchange of atmospheric $CO_2$. But large uncertainties surround the impacts of land-use change emissions, climate warming, the frequency of droughts, and $CO_2$ fertilization. This culminates in poorly quantified carbon stocks and carbon fluxes even for the major ecosystems of Africa
5  (savannas, and tropical evergreen forests). Contributors to this uncertainty are the sparsity of (micro-)meteorological observations across Africa's vast land area, a lack of sufficient ground-based observation networks and validation data for $CO_2$, and incomplete representation of important processes in numerical models. In this study, we therefore turn to two remotely-sensed vegetation products that have shown to correlate highly with Gross Primary Production (GPP): Sun-Induced Fluorescence (SIF) and Near-Infrared Reflectance of vegetation (NIRv). The former is available from an updated product that we recently
10  published (SIFTER v2), which specifically improves retrievals in tropical environments.

A comparison against flux tower observations of daytime-partitioned Net Ecosystem Exchange from six major biomes in Africa shows that SIF and NIRv reproduce the seasonal patterns of GPP well, resulting in correlation coefficients of >0.9 (N=12 months, 4 sites) over savannas in the northern and southern hemispheres. These coefficients are slightly higher than for the widely used MPI-BGC GPP products and Enhanced Vegetation Index (EVI). Similar to SIF signals in the neighboring
15  Amazon, peak productivity occurs in the wet season coinciding with peak soil moisture, and is followed by an initial decline during the early dry season, that reverses when light availability peaks. This suggests similar leaf dynamics are at play. Spatially, SIF and NIRv show a strong linear relation (R >0.9, N=250+ pixels) with multi-year MPI-BGC GPP even within single biomes. Both MPI-BGC GPP and EVI show saturation relative to peak NIRv and SIF signals during high productivity months, which suggests that GPP in the most productive regions of Africa might be larger than suggested.



# 1 Introduction

Gross Primary Production (GPP) is the carbon dioxide ($CO_2$) flux between the terrestrial biosphere and the atmosphere by the terrestrial plants via plant photosynthesis, and it is the largest $CO_2$ flux on the planet (Beer et al., 2010). In determining African Net Ecosystem Exchange (NEE), GPP was more important than total ecosystem respiration (TER) (Ciais et al., 2011; Ardö, 2015). It dominates the interannual variability of the terrestrial ecosystem carbon uptake and as a consequence of fertilization, it is likely to continue its substantial increase and play an important role in the carbon-climate coupling (Vermote et al., 1997; Friedlingstein et al., 2019). Therefore, quantification of the spatiotemporal variations in GPP is important to assess biogeochemical cycling in the terrestrial biosphere, ecosystem functioning, carbon budgets, and food production in the context of global climate change. Accurate quantification of GPP is still a challenge at scales beyond that of a single ecosystem level, due to the lack of a reliable GPP signal that can be observed worldwide. Especially in the highly productive tropical regions, the lack of both large-scale GPP signals and local measurements leads to a lack of understanding of how environmental changes drives carbon exchange. As a result, we can only crudely describe the carbon balance of these regions in the current and future climate.

For example, it is still unclear whether African biomes are a net sink or source of atmospheric $CO_2$, and there is generally low confidence in the simulated climate change response of the region in Earth System Models (Williams et al., 2007; Ciais et al., 2009). Africa has a significant and growing role in the global carbon budget, and it is likely that a sizeable fraction of the observed interannual variability of the global carbon cycle (Cox et al., 2013; Ballantyne et al., 2018) can be attributed to the African continent (Williams et al., 2007). Despite its global and regional importance, Africa has few environmental observations networks (Fisher et al., 2013) leaving so-called global atmospheric $CO_2$ inversions (Peters-Lidard et al., 2007; Peylin et al., 2013; Gaubert et al., 2019) poorly constrained. This leaves Africa as the most uncertain, and error-prone continent for carbon flux estimates.

Recently, Palmer et al. (2019) suggested that tropical Africa is an unexpectedly large net source of $CO_2$ to the atmosphere, reaching nearly 1.5 PgC/yr during the 2015/2016 El Niño event. According to two separate satellite-products that retrieve column integrated $CO_2$ ($XCO_2$) from observed radiances, the northern part of Africa contributes most to the this carbon source (Mengistu and Mengistu Tsidu, 2020). Hotspots of emissions in the Congo basis and western Ethiopia, tentatively associated with land use changes over lands with high soil carbon densities, are partly responsible for this source. An important next step is to verify these finding independently using ground-based measurements of $CO_2$ fluxes (GPP, TER, and NEE) and $CO_2$ mole fractions, as $XCO_2$ retrievals are still in a development phase, and previous versions of these products have displayed various biases despite enormous efforts and great diligence from the retrieval experts (O'Dell et al., 2018).

Arguably the most reliable measurements of NEE come from the eddy-covariance technique (Baldocchi et al., 2001). However, there are still uncertainties in the partitioning of the measured net ecosystem exchange flux into GPP and respiration (Reichstein et al., 2005; Lasslop et al., 2012). Furthermore, the eddy-covariance methods only provide measurements over a restricted area covered by their observation footprints with size and shape that vary with tower height, canopy physical characteristics and wind velocity and by the limited and biased spatial distribution of towers across the globe (Schimel et al., 2015).



In Africa there are relatively few eddy-covariance measurement sites, and the data from these towers often suffer from gaps in their observational records. On the other side, terrestrial and ecosystem models can simulate GPP over a varied spatial and temporal scales all over the globe, but the reliability of such calculations heavily depends on both the input data and the model formulation, which often are not specific for African (or Tropical) biomes. For example, Fisher et al. (2013) estimated average
GPP of the African tropical forest to range from 1.4 to 4.0 kgC m$^{-2}$ yr$^{-1}$, indicating large variability among nine global dynamic vegetation models.

The seasonal dynamics of GPP over tropical ecosystems has been discussed widely due to contrasting observations from remote-sensing and eddy-covariance platforms over the south American Amazon basin (see Restrepo-Coupe et al. (2013) and references therein). In addition to photosynthetic active radiation (PAR) and vapor pressure deficit (VPD), there is a clear
contribution of soil moisture is stress for short vegetation, and for the changing photosynthetic capacity of leaves as a function of age in broadleaf vegetation (Xiao et al., 2006; Huete et al., 2006), in shaping the GPP seasonal cycle. In seasonally wet forests, GPP typically peaks in the wet season when VPD is low and soil moisture high, and declines in the early dry season only to increase again well before the rainfall minimum, as freshly grown leaves take advantage of the maximum in PAR (Lopes et al., 2016). Areas with low vegetation (shrubs, grasses, sparse trees) instead show a decline of GPP throughout the
dry season, as soil moisture and high VPD limit productivity of the vegetation. These patterns were confirmed in three separate studies using remote-sensing observations of sun-induced fluorescence from GOSAT (Lee et al., 2013), GOME-2A (Koren et al., 2018), and TROPOMI (Doughty et al., 2019) over the Amazon.

Similar to SIF, the near-infrared of vegetation (NIRv), which is the product of the normalized vegetation index (NDVI) and total scene near infrared reflectance (NIRT), was found to provide a good proxy for GPP (Badgley et al., 2017; Turner et al.,
2019; Baldocchi et al., 2020). Therefore, we use the Sun-Induced Fluorescence of Terrestrial Ecosystems Retrieval (SIFTER, van Schaik et al., 2020) data from the GOME-2A instrument and NIRv from MODIS to assess the usefulness of these signals to capture the seasonality and magnitudes of GPP derived from six eddy covariance flux towers from Africa in the overlap years between the years 2007-2014. We also test the robsutness of SIF and NIRv to track the seasonality of GPP for the major biomes in comparison to the widely used machine-learning approach of MPI-BGC GPP, as well as to other vegetation remote-sensing
indices like NDVI (Kong et al., 2016) and EVI (Arvor et al., 2011). Further, we assess the relationship between the satellite observations and (model-generated) soil moisture (SM) and incoming shortwave (SWR) radiation in the region. Finally, we derived a plant-functional specific linear relation between eddy-covariance GPP and SIF/NIRv, to quantify integrated GPP from remotely-sensed signals.

## 2 Data and Methods

### 2.1 Study area


The relationships between NIRv, SIF, and GPP were studied for the major biomes over Africa. The dominant biomes of the region are: broad-leaf evergreen forest (BLEF), C3 grasses (C3), shrubs (Sh), and C4 grasses (C4) (Fig. 1e). Northern and southern parts of Africa operated in anti-phase in their summer insolation, precipitation and other environmental stresses.





Therefore, we subdivided the shrubs and C4-grasses into a Northern Hemisphere (NH) part and Southern Hemisphere (SH) part. However, we didn't split up the BLEF which is at the tropical rainbelt and shows weak symmetry between the north and south of the equator as well the C3 due to its smaller coverage over the northern part of Africa. The vegetation type distribution is based on the terrestrial biosphere model SiBCASA (Schaefer et al., 2008; van Schaik et al., 2018).

The seasonal movement of the Inter-Tropical Convergence Zone (ITCZ) drives the climate of Africa in response to changes in the location of maximum solar heating in the region. The ITCZ seasonally migrates north and south of the equator as the latitude of maximum solar insolation varies, causing equatorial Africa to be characterized by double rainfall maximum rainy seasons (Singarayer et al., 2017). However, in East Africa, local topography leads to spatially variable temperatures and a complex distribution of rainfall (Gebrechorkos et al., 2019). The seasonal dynamics of vegetation are strongly controlled by these

climatic conditions (Stephenson, 1990) through the key processes of photosynthesis, respiration, and transpiration. Mainly, the length of the dry season has often been emphasized as a major factor controlling the vegetation structure and patterns in the tropics (Ngomanda et al., 2009; Vincens et al., 2007). November - April are the wettest, whereas June - September are the driest months for the regions with C3, and Sh-SH biomes (Fig. 1d). On the other hand, July - September are the wettest, while December to March are the driest months for regions of Sh-NH and C4-NH. For BLEF regions the precipitation is higher than

100 mm/month throughout the year.

## 2.2 Gridded data sets

In this study we use Level 3 GOME-2 Sun-Induced Fluorescence of Terrestrial Ecosystems Retrieval SIF (v2.0) at 737 nm monthly data at a spatial resolution of $0.5° \times 0.5°$ covering the period from 2007 to 2016. For a fair comparability of the SIF with vegetation indices we normalized SIF by the cosine of the solar zenith angle. We used the KNMI/WUR SIF retrieval,

which is particularly suited for tropical conditions (van Schaik et al., 2020; Koren et al., 2018). The retrieval code uses a much larger dataset to construct the reference atmospheric spectra used to distinguish the small SIF signals from the complex structure of transmittance and reflectance from other atmospheric constitutes such as water vapor (van Schaik et al., 2020).

The NIRv represents the fraction of reflected Near-Infrared Reflectance (NIR) of light that originates from vegetation. NIRv was first described as a proxy for photosynthesis by Badgley et al. (2017). To obtain more accurate retrievals of the contribution

of vegetation to observed NIR reflectance under a wide array of field conditions, including over sparse canopies and regardless of soil brightness, recent studies use NIRv for vegetation productivity and computing a NIRv-based meddled SIF at 760 nm (Badgley et al., 2017; Zeng et al., 2019). In this study we used NIRv at a spatial resolution of $0.5° \times 0.5°$ and a monthly temporal resolution for the years 2007-2016. We also used NIRv at a higher resolution ($0.05° \times 0.05°$, daily) in a comparison with flux tower GPP. NIRv data used here were calculated using surface reflectance data from MODIS collection MCD43C4

v006 (Schaaf and Wang, 2015) from 2007 to 2016 as:

$$NIRv = \rho_{nir} \times \left( \frac{\rho_{nir} - \rho_{red}}{\rho_{nir} + \rho_{red}} - 0.08 \right) \tag{1}$$

where $\rho_{nir}$, $\rho_{red}$ are reflectances acquired in the near infrared (841–876 nm) and red (620–670 nm) portions of the electromagnetic spectrum respectively (Huete et al., 2002). A constant 0.08 is subtracted to reduce the effects of the bare soil



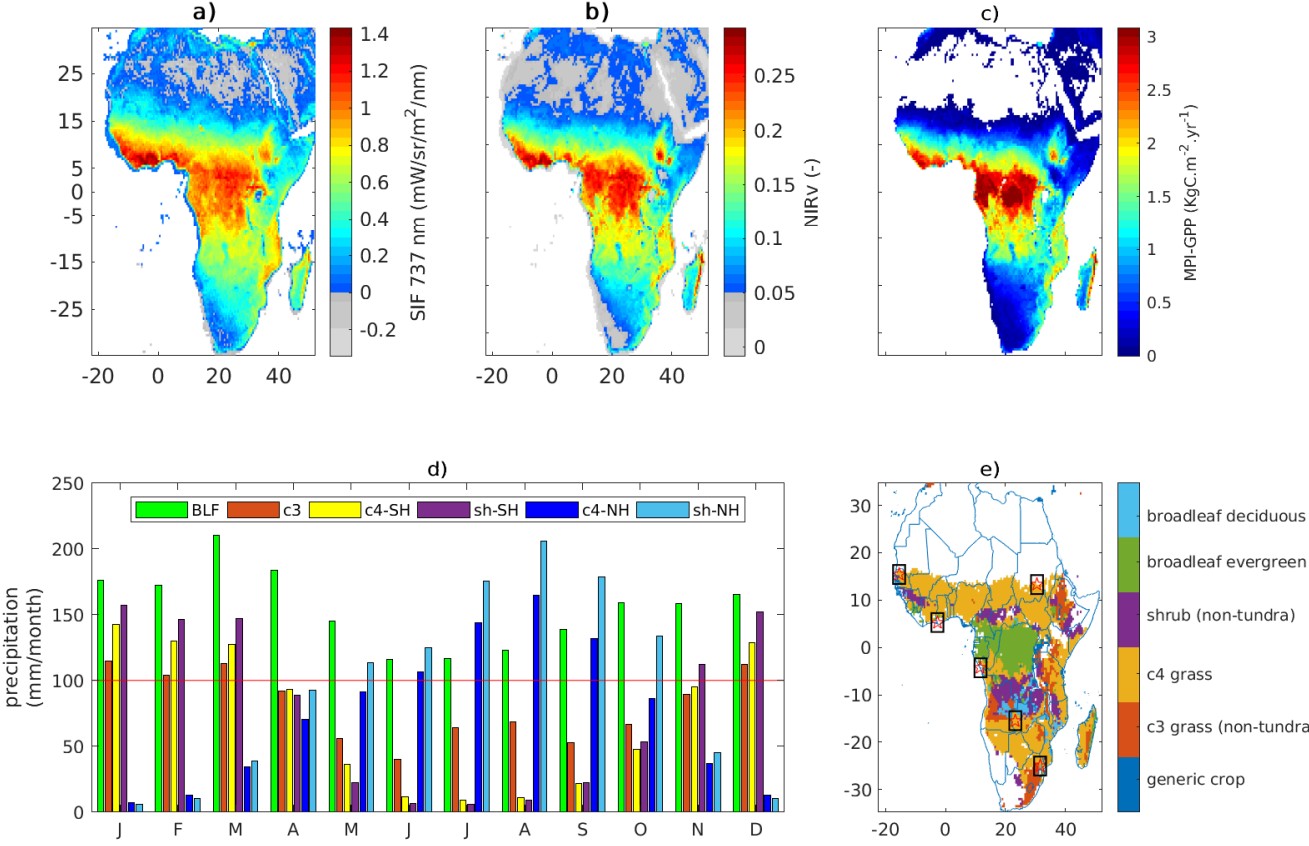

**Figure 1.** Annual mean SIF (a) and NIRv (b) averaged over the years 2007 – 2016 at 0.5°×0.5° resolution. (c) Climatology of gross primary production (Beer et al., 2010) at 0.5°×0.5° resolution. (d) Monthly average precipitation computed from the GPCC product covering the period from 1951 to 2000 for the major biomes of Africa. The red line indicates a reference of 100 mm/month. (e) Vegetation types from the terrestrial biosphere model SiBCASA. The white color refers to none vegetated regions. The rectangular window with a pentagon star at the center shows the distribution of flux towers in Africa.

(Baldocchi et al., 2001; Huang et al., 2019). Uncertainties in NIRv are largely due to in accuracy in measurements of canopy architecture, including the leaf projection function and the clumping index, both strongly vary in time and space (Zeng et al., 2019).

Monthly global ecosystem estimation of terrestrial GPP was made available by the Max Planck Institute's Biogeochemical
5  Integration Group (Jung et al., 2011). This GPP estimation is constructed using a machine learning method to upscale information from flux-towers up to a 0.5°×0.5° grid, aided by gridded meteorological and remote sensing co-variables covering the period from 2007 to 2011 was used hereafter ensemble GPP (MPI-BGC GPP). This product is available at a monthly temporal



resolution (downloaded from https://www.bgc-jena.mpg.de/geodb/projects/Data.php) covering the period from 2007 to 2011.

Enhanced Vegetation Index (EVI) and the Normalized Difference Vegetation Index (NDVI) are the two most widely used vegetation indices for monitoring vegetation conditions and have significant relationships with GPP (Xiao et al., 2005). Monthly

EVI and NDVI from Moderate Resolution Imaging Spectroradiometer (MODIS) collection of MOD13C2 at a a spatial resolution of $0.05° \times 0.05°$ in the years 2007-2015 was used in the study. Compared with NDVI, EVI is less sensitive to soil background variations and remains sensitive over dense vegetation (Huete et al., 2002). For that reason we focus on comparison with the EVI than with NDVI.

Monthly precipitation data from Global Precipitation Climatology Centre (GPCC) at $0.5° \times 0.5°$ spatial resolution was em-

ployed to show the dry and wet months of the region for each major biomes. The GPCC Full Data Monthly Product Version 2018 covers the period from 1891 to 2016; this new extended product version using the new GPCC climatology as analysis background was generated in May 2018 and can be accessed from http://gpcc.dwd.de. In addition we used monthly temperature data from European Centre for Medium-Range Weather Forecasts (ECMWF) atmospheric reanalysis ERA-Interim (with a $0.5° \times 0.5°$ grid). Furthermore, we used monthly soil moisture (SM) and incoming downward shortwave radiation (SWR)

from the Global Land Data Assimilation System Version 2.1 (GLDAS2.1) of National Aeronautics and Space Administration (NASA) Goddard Space Flight Center (GSFC) at a spatial resolution of $0.25° \times 0.25°$ covering the period from years 2007-2016 (Peters-Lidard et al., 2007).

## 2.3 Flux tower data

Standardized eddy covariance flux data are available under the fair-use data policy of FLUXNET2015 dataset. The data pro-

cessing of FLUXNET2015 dataset ensures inter-comparison and quality assurance and control across sites (Vuichard and Papale, 2015). A collection of eddy covariance flux data from six regions in Africa was used in this study to assess the correlation between SIF, NIRv and GPP at ecosystem level. Specifically, available monthly GPP products from daytime partitioning of fluxes (GPP_DT_VUT_REF) from Tire 2 FLUXNET2015 synthesis www.fluxnet.org. These variables were screened using quality flags so that only samples that are either measured (flag = 0) or good quality (flag = 1) were retained. An overview of

the selected towers is given in Table 1. In addition to these six African flux towers we used data from Brazil BR-Sa1 flux tower for a better representation for GPP in broadleaf evergreen forests.

## 2.4 Analysis Method

To compare the gridded data sets (SIF, NIRv, MPI-GPP) with the GPP measurements from flux towers, we extracted data from a $4° \times 4°$ window surrounding each flux tower. The flux towers have a footprint of about 1 km$^2$ and it is hard to

compare them to areas that are 200 km$^2$ centering the tower which includes many vegetation types. However, we use the vegetation mask to exclude grid cells with different vegetation from the tower's vegetation; this will account for the land heterogeneity of the regions. The variation in climate condition was addressed by splitting up the shrubs and C4 grasses over the Northern and Southern Hemispheres. The empirical relationship between SIF and GPP is complicated by the fact that we





**Table 1.** Information on flux-tower sites in Africa and Brazil. The symbol '*' indicates that flux observations before 2007 were not used in this study. The major biomes are savanna (SAV), evergreen broadleaf forest (EBF) and decidious broadleaf forest (DBF). The distribution of the towers over Africa is indicated in Fig. 1e. Note that we also included the details of one tower outside of Africa, BR-Sa1, that was used for our analysis of tropical evergreen broadleaf forests.

| Site ID | Site name | Country | Vegetation | Latitude (°N) | Longitude (°E) | Start date | End date |
|---------|-----------|---------|------------|---------------|----------------|------------|----------|
| CG-Tch | Tchizalamou | Congo | SAV | -4.289 | 11.656 | 01/2006* | 12/2009 |
| GH-Ank | Ankasa | Ghana | EBF | 5.26854 | -2.69421 | 01/2011 | 12/2014 |
| SD-Dem | Demokeya | Sudan | SAV | 13.2829 | 30.4783 | 01/2005* | 12/2009 |
| SN-Dhr | Dahra | Senegal | SAV | 15.40278 | -15.43222 | 01/2010 | 12/2013 |
| ZA-Kru | Skukuza | South Africa | SAV | -25.0197 | 31.4969 | 01/2000* | 12/2013 |
| ZM-Mon | Mongu | Zambia | DBF | -15.43778 | 23.25278 | 01/2000* | 12/2009 |
| BR-Sa1 | Santarem-Km67 | Brazil | EBF | -2.86 | -54.96 | 01/2000* | 12/2011 |

only observe a fraction of emitted SIF photons and that this fraction depends on the direction of observation (Porcar-Castell et al., 2014). Even if the mechanistic link between remotely sensed vegetation reflectance and GPP is complex (Porcar-Castell et al., 2014), Guanter et al. (2014) and Sun et al. (2017) showed that a simple linear relationship between SIF and tower based GPP as reasonably convenient framework for presenting and evaluating arguments and counterarguments for the SIF-GPP relationship. Here, we predict GPP from these remotely sensed vegetation reflectance using a linear regression between GPP and these signals as:

$$y = ax + b \tag{2}$$

where $y$ is the GPP obtained from SIF or NIRv signals, $x$ is the SIF or NIRv signal and $a$ and $b$ are the slope and y-intercept of the fitting line, respectively. The conversion of SIF and NIRv to GPP is achieved by applying these fitting to all monthly SIF and NIRv values for each vegetation type separately.

The linear relationship between SIF and GPP in Eq. 2 may be rationalized with the formulation based on the concept of light use efficiency (LUE) (Monteith, 1972) in a simple parametric LUE-model GPP as follows:

$$GPP = APAR \times LUE_p \tag{3}$$

where APAR is the absorbed photosynthetically active radiation expressed in radiance units and $LUE_p$ is the light use efficiency of photosynthesis, which represents the efficiency of energy conversion for gross $CO_2$ assimilation. Similarly, SIF can be expressed as:

$$SIF = APAR \times LUE_f \tag{4}$$

where $LUE_f$ is the effective light use efficiency of canopy fluorescence, which accounts for both the fluorescence yield and the fraction of emitted photons escaping the canopy (Damm et al., 2015). These equations can be combined making the





dependence on light implicit,

$$GPP \approx SIF \times \frac{LUE_p}{LUE_f} \tag{5}$$

SIF has negative values due to noise in its retrieval and zero value of SIF may not result in a zero value, thereby we do not force the regression to pass through the origin and as there will be intercepts 'b' as in Eq. 5. Further, linear relationship

between NIRv and SIF can be rationalized by the fact that both SIF and NIRv are jointly dependent on the flux of the fractional interceptance of vegetation, incoming solar radiation, and the fraction of photons that escape from the canopy, these two are strongly related measurable fluxes (Zeng et al., 2019).

## 3 Results

### 3.1 GPP proxies and eddy-covariance derived GPP estimates

Spatial patterns in climatological NIRv, SIF, and MPI-BGC GPP are very similar across large scales, with maximum annual mean productivity in tropical broadleaf forests. Fig. 1a-c shows that productivity changes strongly at the borders of the plant-functional types, which is interesting because only the MPI-BGC product was actually informed by a PFT-map in its machine-learning, while the satellite observations provide independent spatial views on productivity. Both products suggest additional variations within PFTs not represented in the MPI-BGC map, as would be expected based on the higher volume of observed

data in the remote-sensing products.

NIRv, SIF, EVI and MPI-BGC GPP generally capture seasonal patterns of tower GPP well, except at the Ghana GH-Ank flux tower where from all products only SIF yields the expected positive correlation with eddy-covariance derived GPP observations. Fig. 2 shows the observation-derived and simulated seasonal cycles of GPP, and the generally high (R>0.8) seasonal correlations. SIF shows more rapid changes in signal during the transitions from wet-to-dry periods than other proxies.

The May-June-July period at the savannah site CG-Tch is an example, and indicates that SIF responds more rapidly to the decline of photosynthesis in wilting grasses, which are still green and reflective enough to affect NDVI, EVI, and NIRv. Such a more immediate response of SIF to water stress was also observed by others (Chen et al., 2019; Tian et al., 2020).

SIF and NIRv have a higher monthly correlation with the EC-GPP at most of the towers than EVI and MPI-BGC GPP (Supplementary Fig. S1). Luus et al. (2017) also found this, and suggest this is because chlorophyll content seen through

NDVI and EVI adapts slowly to stress and it can take weeks for leaves to lose their green color (Hew et al., 1969). EVI and NDVI not only had generally much weaker correlations, but also show saturation when GPP is high (Supplementary Fig. S1). SIF/NIRv had strongest correlations (R > 0.90) with the EC-GPP in C4 grass vegetation sites (SD-Dem and SN-Dhr) sites, while weak or no relationship was found for broadleaf evergreen vegetation (GH-Ank). This lack of relation was also found previously by (Li et al., 2018) over rainforest regions, and is further discussed in Section 4 . Even over evergreen forests where

GPP is high throughout the season (see Fig. 1c) NIRv and SIF tracks seasonality of GPP well.

For a more detailed look, we also compare EC-GPP to daily NIRv signals, derived from high-resolution ($0.05° \times 0.05°$) MODIS radiances instead of coarse averaged MODIS NIRv ($0.5° \times 0.5°$). At this daily timescale, we again find a very strong



**Figure 2.** The skill of NIRv, SIF, EVI and MPI GPP in capturing the seasonal cycle of GPP from flux towers. The shaded area indicates the standard deviation around the monthly means. Note that NIRv, SIF and EVI do not have the same unit as GPP and their values are provided on the secondary y-axis.

correlation over Northern Africa, while this correlation decreases for the equatorial sites (Fig. 3). And again, we see a weak correlation at GH-Ank, Ghana (Fig. 3 Gh-Ank). The high resolution NIRv mostly improves the comparison for sites with more heterogeneous vegetation cover (GH-Ank and ZM-Mon, ZA-kru), whereas no significant improvement for less heterogeneous sites (SN-Dhr and SD-Dem). In contrast to the low climatology correlation for tropical evergreen broadleaf forest (GH-Ank, R = -0.44), the correlation in the interannual variation of NIRv and GPP is higher (R = 0.21) (See supplementary Table S1). These results are illustrative for tropical ecosystems, where GPP variations are irregular and strongly coupled to photosynthetic capacity changes of vegetation (Restrepo-Coupe et al., 2013, 2017).



**Figure 3.** Comparison of coarse NIRv at $0.5° \times 0.5°$ (left panels) and high resolution NIRv at $0.05° \times 0.05°$ (right panels) with flux tower measured GPP (EC-GPP) for a selection of three African flux towers (GH-Ank, SD-Dem and ZM-Mon). The temporal correlation between NIRv and EC-GPP is given in each panel.

## 3.2 GPP proxies across the major biomes of Africa

We next extend this view from the level of individual towers, to the scale of six major biomes in Africa (Fig. 1e) by spatially averaging our productivity products. Also then, wet months show higher SIF and NIRv values than dry months for all biomes of the region. Both SIF and NIRv shows a strong linear relation with that of the MPI-BGC GPP estimates. Signals from both SIF

5  and NIRv were correlated well with MPI-BGC GPP over these biomes with a correlation of $R^2 > 0.85$ for NIRv and $R^2 > 0.77$ for SIF. However, the correlation was moderate for broad leaf evergreen forests with a correlation of 0.38 for NIRv and 0.16 for SIF (see supplementary Table S2). This seasonality shown in Fig. 4 agrees closely with that seen for similar biomes in the Amazon (Girardin et al., 2016; Koren et al., 2018) and confirms the known strong water-control over GPP in tropical vegetation (Abdi et al., 2017; Bonal et al., 2016). Correlations with shortwave radiation are thus strongly negative, especially over short



vegetation like shrubs and grasses. Over evergreen forests, SIF and NIRv shows a double-peaked seasonality and a decrease of productivity during the dry seasons despite high SWR and high leaf area index (see Fig. 4 Broadleaf evergreen), suggesting an influence of photosynthetic capacity on GPP that has been noted before to be not yet represented by most biosphere models and light use efficiency models (Bhattacharya, 2018). Clouds can strongly reduce direct solar radiation during the wet season
which increases the ratio of diffuse versus direct solar radiation, possibly increasing productivity (Hollinger et al., 1999).

Mostly, EVI and MPI-GPP closely agree on the satellite-observed seasonality at these larger scales, but EVI appears late in simulating the wet-dry season (Sep-Dec) decline in signal for C3 grasses and shrubs (see SIF-vs-EVI hysteresis plots in Supplementary info Fig. S3). This delayed response resembles that described by Luus et al. (2017) for high-latitude short vegetation, which greened up in EVI 9 days before spring photosynthesis started. Our response is opposite in the sense that we
see photosynthesis decline already before the seasonal brown down of the Savannah. To further investigate this, we therefore turn to the main drivers, water and light, of the African seasonal cycle in GPP.

The biome-integrated productivity in Africa is seasonally strongly controlled by soil moisture, with a weak influence of light availability superimposed. In Fig. 5 this is recognized by the positive correlation between SIF/NIRv (which independently display the exact same patterns) and soil moisture in both hemispheres. Peak productivity coincides with peak soil moisture
that occurs in September on the NH, and in March on the SH. Interestingly, the lead up to peak productivity occurs more slowly than its subsequent decline even at the same soil moisture levels, evident when comparing the pairs of months (8,10) and (7,11) for the NH, or pairs (1,4) and (12,5) for the SH. Translating these points to the SWR diagrams in panels (a) and (c), a notable difference between the hemispheres appears: in the SH peak productivity occurs while light-availability continues to decline, creating the elliptical shape of the hysteris diagram. In the NH however, peak productivity happens at minimum
light-availability becoming larger at the same soil moisture levels past the peak productivity.

### 3.3 SIF-GPP/NIRv-GPP Estimation for the Major Biomes

A SIF or NIRv based GPP estimate across each biome compares independently quite well to MPI-BGC estimated GPP. Figure 6 shows the GPP estimated by applying the SIF/NIRV-vs-GPP relation derived at EC-sites to biome-wide satellite observations. We chose the relation between the remotely sensed reflectances and the EC-GPP from Senegal SN-Dhr, Congo CG-Tch, Sudan
SD-Dem and ZA-Kru towers to estimate GPP of the Northern Shrub, the Southern shrub, C4-grass and C3-grass respectively to represent different biomes over the northern and southern Africa (see Supplementary Fig. S1 for the fitting equations applied). Due to the weak relationship between SIF/NIRv with Ghana GH-Ank GPP, a flux tower in tropical rainforest of Africa with a broadleaf evergreen vegetation, we used a linear relation between SIF/NIRv with EC GPP data from Brazil BR-Sa1 flux tower which is also in tropical rainforest region with a broadleaf evergreen vegetation and shows a better relation with these
signals than the GH-Ank site (See Supplementary Fig. S2). The good correspondence to MPI-BGC GPP partly results from the EC-sites being part of the training algorithm for that product (Jiang and Ryu, 2016), but we note that the spatiotemporal drivers in our product (SIF/NIRv) are much different from that in MPI-BGC (PAR, T, PFT, precip, ...). Mostly, it shows that a reasonable first estimate of spatiotemporal GPP patterns can be based on SIF and NIRv without the need for the more complex

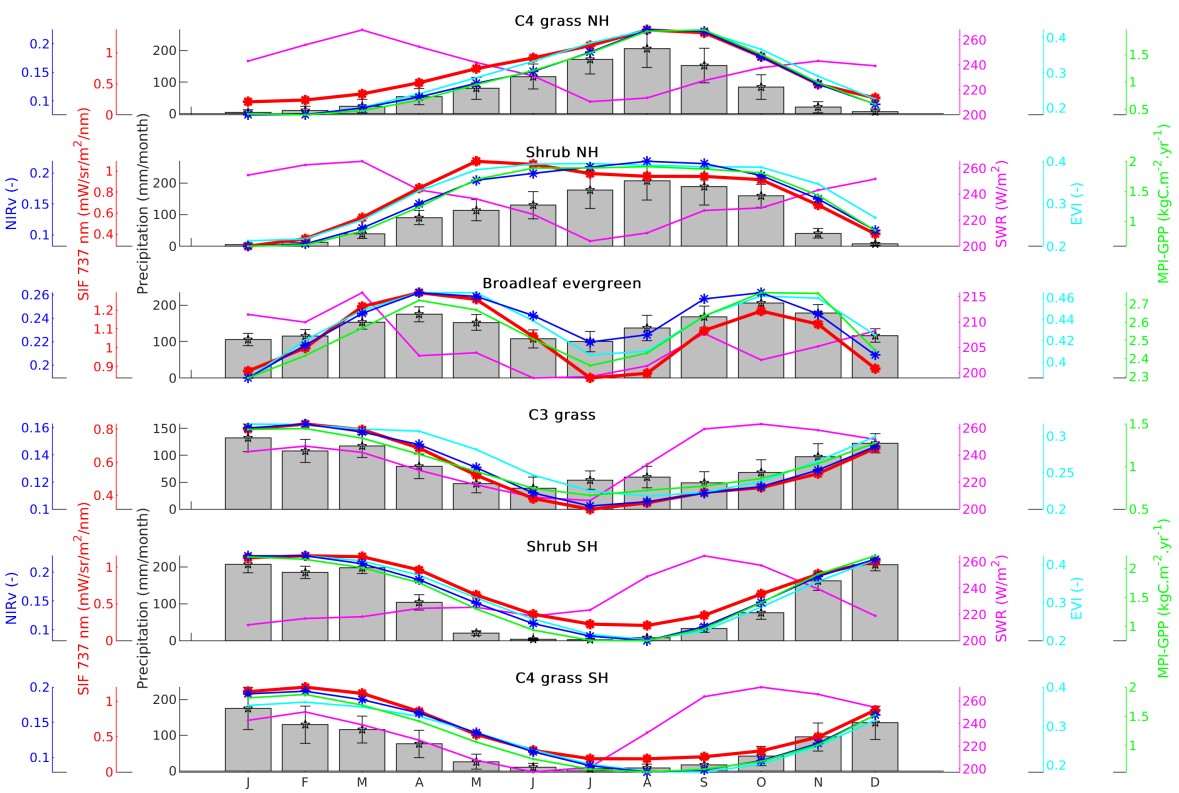

**Figure 4.** Seasonal cycles of NIRv, SIF, precipitation, short wave radiation (SWR), EVI and MPI GPP at $0.5° \times 0.5°$ resolution over major biomes of Africa covering the period from 2007 to 2011. The error bars provided for precipitation denote the standard deviation around the mean.

and data-intensive machine-learning approach. At least, it captures the large differences between the major biomes of Africa, allowing further study of their seasonal dynamics, drought response, and contribution to tropical GPP.

## 4 Discussion

We found the relationship between SIF/NIRv and GPP for croplands to be strongest, with R > 0.90 for C4 vegetation at
5 both the Sudan (SD-Dem) and Senegal (SN-Dhr) sites (Supplementary Fig. S1). A much weaker relation was obtained for broadleaf evergreen vegetation such as in Ghana (GH-Ank). This agrees with Li et al. (2018), who also showed a weak relationship between SIF and GPP over rainforest sites. Previous authors suggested this may result from the inefficiency of satellite measurements to detect the canopy activities of tropical forest (Tang and Dubayah, 2017), due to limitations in their



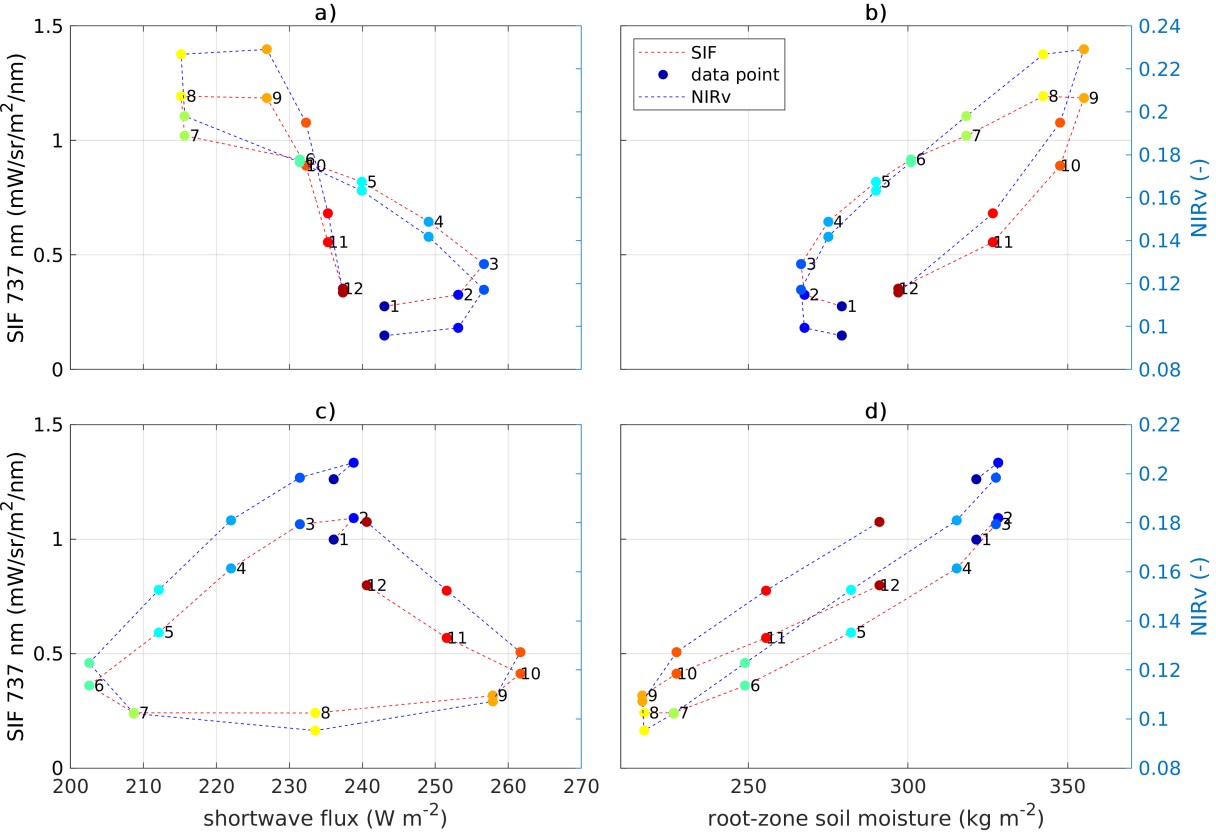

**Figure 5.** Relation between SIF/NIRv with downward shortwave radiation (left panels), root zone soil moisture (right panels) for major African biomes. Upper panels show results over the vegetated regions of Northern Africa (north of the equator) while the bottom shows the results for Southern Africa (south of the equator). The numbers in the plot, and the color of the markers, refer to the month of the year.

retrieval due to atmospheric cloud contamination (Frankenberg et al., 2014; Doughty et al., 2019), or from the EC measurement technique itself (Hayek et al., 2018).

Eddy correlation measurements over rainforests are more complicated than over flat vegetation due to the presence of uneven tall canopies (Mercado et al., 2006) as well as stable atmospheric conditions at night (Miller et al., 2004). This comes on top of the uncertainty incurred on the derived GPP, which requires a partitioning of the measured net ecosystem exchange during turbulent conditions (Reichstein et al., 2005). The methodology used assumes a temperature-dependency of ecosystem respiration to remove its influence during daytime, such that GPP can be determined from the residual of measured NEE and measurement-derived TER. For tropical sites this T-dependency is often assumed negligible (Restrepo-Coupe et al., 2013). The partitioning approach is furthermore not well-tested for tropical ecosystems because of a lack of long data records, the larger



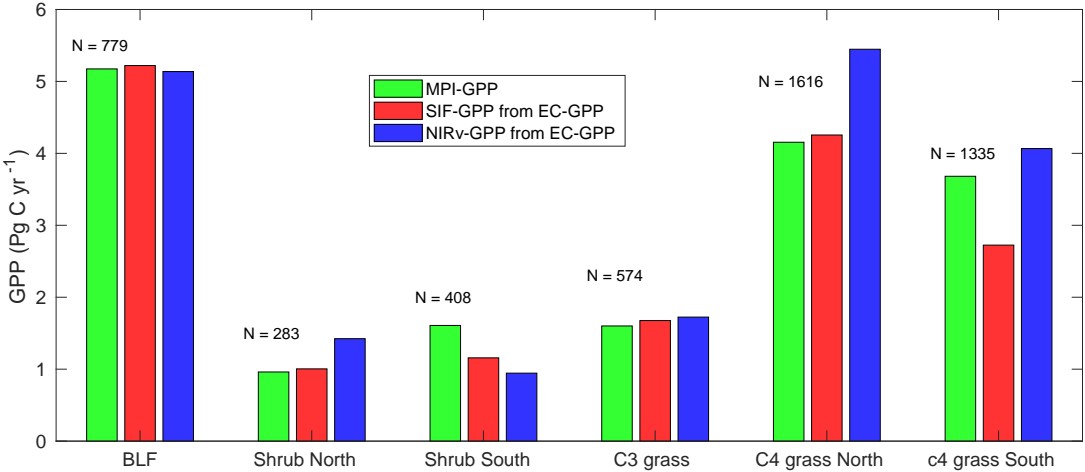

**Figure 6.** Comparison of aggregated MPI-BGC GPP, SIF-GPP and NIRv-GPP for major biomes in Africa. N is the number of grid box of size $0.5° \times 0.5°$ used in the aggregation.

uncertainty in determining nighttime TER (Kruijt et al., 2004), and the often seen nighttime storage that results in peak NEE fluxes in the early morning (Araújo et al., 2002). Finally, tropical TER is likely to experience larger control of temperature and moisture on TER (Chambers et al., 2004).

The tower GPP from the GH-Ank site shows limited seasonality (Fig. 3). GPP over Evergreen Tropical Forest vegetation,
is poorly captured by all products. Where NIRv, EVI, and MPI-BGC all show a phase-lag of nearly 4-months, SIF manifests a double peak structure and much too low annual mean SIF relative to the other datasets. In general, SIF and NIRv were better proxies in capturing the seasonal dynamics of GPP over most sites. The relationships can be improved by using higher resolution products instead of these coarse resolutions. Here we used SIF from the GOME-2 instrument which has a larger footprint ($40 \times 80$ km$^2$) and global gridding with a spatial resolution of $0.5° \times 0.5°$ at monthly time scale. TROPOMI SIF
is a promising alternative for future studies of the African carbon cycle. TROPOMI SIF has a higher spatial resolution than GOME-2 SIF and a more frequent coverage. Global TROPOMI SIF data was first shown by Köhler et al. (2018). Their study also included a detailed view of the North-African Nile Delta. In addition, Doughty et al. (2019) used TROPOMI to study the seasonality of the Amazon, showing the capability of this instrument to capture seasonal dynamics of tropical ecosystems. Moreover, high resolution ($0.05° \times 0.05°$) SIF can be modeled using explanatory variables which are available at both fine and
coarse resolution (Guanter et al., 2014; Zeng et al., 2019). These results are illustrative for tropical ecosystems, where GPP variations are irregular and strongly coupled to leaf phenology of vegetation (Restrepo-Coupe et al., 2013, 2017).

Remote sensing data driven models are widely used for estimating plant photosynthesis and they are linearly dependent on the amount of solar illumination, amount of water content in the soil and plant canopy (Ceccato et al., 2001). Most of these models assume information in Fraction of Absorbed Photosynthetically Active Radiation (fAPAR) and Vapor Pressure Deficit
(VPD) are sufficient to accurately estimate the responses of GPP to drought. However, deficits in soil moisture and their effects



on GPP are not necessarily captured by fAPAR or VPD (Stocker et al., 2019) and result in a large uncertainties in these GPP estimations. Our analysis showed seasonality of soil moisture strongly controls plant productivity with a weak intervention of available shortwave radiation. SIF and NIRv have a more strong correlation coefficient of R ≈ 0.97 with soil moisture over Southern Africa than over Northern Africa (R ≈ 0.76). Despite this strong linear relation, during some months, we observe a

very small difference in SIF/NIRv while the difference in the soil moisture was large. This is partly related with the amount of solar radiation. During saturation, when the soil is very moist, the amount of shortwave radiation significantly impacts productivity, whereas during the growing or end period of growing seasons vegetation production has a strong proportion to soil moisture.

GPP estimation from SIF needs a more complicated modeling approach (Norton et al., 2018; Anav et al., 2015) as it needs
assimilation of SIF into a terrestrial biosphere model to estimate the gross uptake of carbon through photosynthesis. However, we apply a simple linear relation between SIF/NIRv and EC GPP and showed reasonably well estimate of GPP over different biomes (Fig.6) (see also supplementary Table S3 for biome specific estimation of GPP per unit area as a response of the major biomes of Africa). The GPP obtained in this simple upscaling method was integrated for each biome found in good consistency with MPI-BGC GPP. SIF-GPP was found more consistent with MPI-BGC GPP than the NIRv-GPP. The NIRv-GPP up scaled
was 31% higher compared to MPI-BGC GPP for C4 grass over Northern Africa. In contrast, it is lower by 42% for shrub over Southern Africa. Guanter et al. (2014) found that the MPI-BGC GPP underestimated the global crop production by 50-70% than SIF-GPP obtained by a fitting to flux towers based GPP from US and Europe crop and grass lands however in African grass land we found SIF-GPP is relatively lower than MIP-BGC GPP.

So far we have focused on remote sensing proxies for GPP, but there are also alternative tracers of GPP. This includes
carbonyl sulfide (COS) which diffuses through stomata, similar to $CO_2$, and is taken up inside leaves (Montzka et al., 2007). The leaf relative uptake (LRU) of COS and $CO_2$ can be used to infer GPP from the COS uptake (Campbell et al., 2008). In addition, the oxygen isotopic composition of atmospheric $CO_2$ ($\delta^{17}O$ and $\delta^{18}O$) can also provide constraints on GPP. The isotopic composition of $CO_2$ is controlled by exchange with leaf water inside plants and the magnitude of this exchange is related to GPP. In particular, $\Delta^{17}O$ (also known as the $^{17}O$-excess, approximately equal to $\delta^{17}O - 0.5 \times \delta^{18}O$) is a promising
tracer because it is less dependent on the water cycle than the more traditional tracer $\delta^{18}O$, and easier to interpret as a tracer for GPP (Hoag et al., 2005). At the leaf and branch-scales, these tracers (COS and $\Delta^{17}O$ in $CO_2$) can be used to estimate GPP (Kooijmans et al., 2019; Adnew et al., 2020), whereas higher up in the atmosphere the signals contained in these tracers represent larger land areas (Koren et al., 2019; Ma et al., 2020), such as the African biomes that we studied here. Whereas SIF and NIRv allow us to diagnose instantaneous productivity, the signals carried by atmospheric tracers contain information on
longer time scales. When sufficient observations become available, these tracers have the potential to provide an additional, independent constraint on productivity across Africa.





## 5 Conclusions

There is substantial uncertainty in GPP estimated by terretrial biosphere models, especially for tropical regions. Particularly, in regions like Africa having very few and sparse observation networks. Thus, the use of satellite fluorescence is highly valuable to complete our understanding of the global and regional carbon cycle. The mean climatology of SIF and NIRv agrees widely with

the MPI-BGC GPP products in large parts of Africa, confirming their values as a more robust GPP proxy than the commonly used MODIS vegetation indices. Comparing SIF and NIRv with flux tower measurements from six EC flux sites in Africa, we found that SIF and NIRv can capture the seasonality of measured GPP over most sites. The relationship between SIF/NIRv and GPP was stronger (R > 0.90) in C4 vegetation examined at both Sudan (SD-Dem) and Senegal (SN-Dhr) sites. SIF and NIRv were found to capture the seasonal cycle well while MPI-BGC GPP products and vegetation index shows a saturation

when production is high. A weak relationship was found for broadleaf evergreen vegetation that was examined in Ghana (GH-Ank). The tower GPP in the GH-Ank site hardly shows the presence of seasonality in GPP. In contrast, both MPI-BGC GPP and satellite fluorescence showed there is a clear seasonality in tropical rainforest GPP that follows the rainfall pattern of the region. Large uncertainties in GPP measurement from the eddy covariance technique in tropical forests may contribute to the weak relation. The correlation that we find for the seasonal cycle of GPP and NIRv for tropical evergreen broadleaf forest

GH-Ank is R = -0.44, whereas the agreement in the interannual variation of NIRv and GPP is higher (R = 0.28 for NIRv extracted from a $0.05° \times 0.05°$ grid). These results are illustrative for tropical ecosystems, where GPP variations are irregular and strongly coupled to leaf phenology of vegetation. One of the primary advantages of NIRv is that it can be calculated using existing satellite sensors, opening the possibility of producing satellite-derived photosynthesis estimates at the global scale going back decades. This opens the door to directly using satellite measurements of NIRv and SIF to drive ecosystem models

and, ultimately, improve our understanding of the physiological and evolutionary processes that govern whole-plant resource use.





*Acknowledgements.* The authors acknowledge NOAA Earth System Research Laboratory, Hadley Centre, Global Precipitation Climatology Centre, Global Land Data Assimilation, MODIS dataset and ECMWF for the data products. The first author also acknowledges Addis Ababa University, Addis Ababa Science and Technology University, Coimbra scholarship group, and University of Groningen for their support through fellowship and access to the research facilities. Data was processed on Cartesius (SURFsara) using a grant for computing time

5   (SH-312-14) from the Netherlands Organization for Scientific Research (NWO). WP and GK acknowledge funding from the ERC project ASICA (GA #649087).





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
