# Peer review of "Supporting information for Sun-induced Fluorescence and Near Infrared Reflectance of vegetation track the seasonal dynamics of gross primary production over Africa"

_Biogeosciences, 2020_

## Referee Comment (RC1) · Anonymous Referee #1 · 3 Sep 2020

The authors compare SIF and NIRv, along with a handful of vegetation indices, against six flux towers located across the African continent. They use these data to build a linear model of SIF and NIRv to estimate GPP across the continent.

I have concerns about the spatial mismatch between eddy covariance measurements and the satellite products used for upscaling. The authors use 0.5 degree satellite imagery and take the further step of aggregating up to 4 degrees (filtering 0.5 degree pixels by dominant land cover type). These average observations are then compared against EC-derived estimates of GPP. Figure 3 suggests that this spatial aggregation

significantly influences the temporal correlation between the satellite measurements and GPP estimates. In the case of GH-Ank, the 0.5 degree measurements of NIRv are dramatically different from the 0.05 data (e.g., the 0.05 degree data show much more temporal variability). For ZM-Mon, the shape of the NIRv curve is quite different during the middle of the growing season when comparing 0.5 to 0.05 degree imagery.

This is a fairly challenging problem to get around. On the one hand, the authors offer a nice proof of concept that SIF and NIRv can be scaled to GPP using continental-scale observations. On the other, higher resolution measurements of SIF are rapidly becoming available (e.g., TROPOMI, as the authors mention) and are already available for NIRv. In fact, a more extensive, global scale analysis of the NIRv-GPP relationship, using tower-scale satellite measurements, has been presented elsewhere (Badgley, Anderegg, Berry, & Field 2019).

At a minimum, the authors might consider quantifying how the scaling issue affects their modeled estimates of GPP. They could do this by comparing the coefficients of a model derived from 0.05 NIRv data against the 0.5 degree data.

The authors could also be more descriptive about how they construct their model. How is missing-ness handled? How are clouds screened for? When aggregating to 4 degrees, these details are going to be quite important for understanding how the final satellite signal is constructed.

I appreciated the authors attempt to use their study to draw inferences about the controls on productivity at the continental scale. I think this is the type of framing and analysis that has the potential to make the paper a nice contribution to the literature, as opposed to simply demonstrating that the SIF/NIRv-GPP relationship holds regional. Much of the analysis centered on a discussion on the controls of seasonality in photosynthesis. On P15 L2-8 the authors write: "Our analysis showed seasonality of soil moisture strongly controls plant productivity with a weak intervention of available shortwave radiation...During saturation, when the soil is very moist, the amount

of shortwave radiation significantly impacts productivity, whereas during the growing or end period of growing seasons vegetation production has a strong proportion to soil moisture."

While possibly true, I think this analysis is a little too broad sweeping. Figure 4 shows that broadleaf evergreen forests have a decline in SWR that coincides with declines in precipitation. Personally, I think it would be quite interesting to see if per-pixel anomalies in SIF and/or NIRv track anomalies in SM. I also think that such an analysis would be more informative about mechanism. Aggregating all the data together across biomes, like in Figure 5, has the potential to hide as much as it reveals, given that averages only reflect the most common SIF-precip/SIF-SWR relationship, as opposed to potentially more complex per-biome or per-pixel relationships.

Minor Comments

P2 L19: "so-called" can have a quite negative connotation. Consider removing.

P5 L1-2: "Uncertainties in NIRv are largely due to in accuracy in measurements of canopy architecture, including the leaf projection function and the clumping index, both strongly vary in time and space (Zeng et al., 2019)." I believe that Zeng agues that NIRv carries information about the leaf projection function, as opposed to the leaf projection function causing uncertainty in NIRv measurements.

P7 LUE framework — How appropriate is the LUE framework when you normalize by cosine of solar zenith angle. Doesn't that mean the APAR signal goes away? How should we interpret what is left?

P13 L3 The manuscript does not address uncertainties in the eddy covariance measurements, so seems unnecessary to spend so much time discussing how the approach is uncertain in tropical context.

P15 L19: Again, the paper does not use COS, making this discussion feel a little out of place.

Works cited

Badgley, G., Anderegg, L.D., Berry, J.A. and Field, C.B., 2019. Terrestrial gross primary production: Using NIRV to scale from site to globe. Global change biology, 25(11), pp.3731-3740.

---

## Referee Comment (RC2) · Anonymous Referee #2 · 29 Sep 2020

**1 Overview:**

Review of "*Sun-induced Fluorescence and Near Infrared Reflectance of vegetation track the seasonal dynamics of gross primary production over Africa*" by Mengistu *et al.*

My sincere apologies for the delay in my review.

Mengistu *et al.* present an analysis of SIF, NIRv, EVI, and NDVI over the African con-

tinent. They compare these remote-sensing products to flux towers across multiple ecosystems. They find soil moisture to be the dominant driver for much of their data. They also find both SIF and NIRv do a better job of reproducing GPP than EVI or NDVI. Overall, the study is both interesting and useful. My main comments relate to quantitatively assessing the claim that SIF and NIRv are performing better. The figures are reasonably easy to follow and the text is quite well written. I would recommend minor revisions.

**2 Major comments:**

Both of my main comments relate to the claim that SIF and NIRv are performing better than NDVI and EVI.

2.1 Are SIF and NIRv actually performing demonstrably better?

In the conclusions, the authors state: *"The mean climatology of SIF and NIRv agrees widely with the MPI-BGC GPP products in large parts of Africa, confirming their values as a more robust GPP proxy than the commonly used MODIS vegetation indices.* However, it's not clear to me that 1) agreement with the MPI-BGC GPP estimate is the metric by which to justify that statement and 2) that the differences between some of the products is not within the noise. The latter statement implies to me that SIF and NIRv are both performing comparably well while NDVI and EVI are performing measurably worse. From examination of Figure 2, it's not clear that is the case. I would certainly agree that NIRv are and SIF are performing better than EVI at CG-Tch, but NIRv and EVI seem nearly identical at GH-Ank while SIF seems to show no real correspondence with the EC data.

I'm wondering if it would be possible to set up some hypothesis test to quantify this.

[Figure]

Or maybe it could be done via bootstrapping. For example the authors could randomly sample the different datasets and compare them to the EC data and the MPI-BGC data then report an uncertainty on the $R$ values. Or report the fraction of times the different remote-sensing products performed better than eachother.

**2.2 Are SIF or NIRv capturing those downregulation effects due to water availability?**

As the authors note, the MPI-BGC and other models (e.g., VPRM) include terms for water availability. The authors show that root zone soil moisture is both important for GPP across Africa and strongly coupled to SIF/NIRv. Does this mean that SIF and NIRv are responding to changes in water availability? If so, are they responding in a manner that better approximates GPP than NDVI or EVI? Related to this, does this mean that models using SIF or NIRv would not need a water availability term? Or do they still require one?

The authors mention that the rapid decrease at CG-Tch starting in June is due to water availability, I would be interested to see a timeseries for individual years that show SIF and NIRv tracking this decrease and some measure of water availability. This could be a supplemental figure, but it would be nice to see that it's actually occurring on individual years.

**3 Minor comments:**

**3.1 Wang et al study?**

The authors may want to discuss the Wang et al. (2020; doi:10.1029/2020JG005732) paper. They provide a nice discussion of what these different remote-sensing measures are telling us.

**3.2 Choice of regions?**

In Section 2.1 the authors mention that they split ecosystems across hemispheres (i.e., NH shrubs and SH shrubs). This reviewer is confused about why this decision was made. The precip and insolation will be out of phase, sure, but wouldn't we expect the response to be similar? Or should we expect a different functional relationship?

**3.3 Use of multiple MODIS products?**

I see the authors used MCD43C4 for $NIR_v$ but MOD13C2 for EVI and NDVI (Page 6, Line 5)? I'm curious why they didn't use the same set of reflectances and then compute the various indices in a consistent manner?

**3.4 LUE models**

Not all of the photons emitted will escape the canopy. Shouldn't there be some canopy escape term in the SIF relationship? If I recall, the Dechant et al. (2020; doi:10.1016/j.rse.2020.111733) paper argued that this canopy escape term is where much of the information is coming from (that we're learning something about structure.

**3.5 Flux tower comparison**

Presumably the flux towers can be influenced by different types of vegetation within the footprint of the tower, why do the authors remove those pixels from the satellite data: *"The flux towers have a footprint of about 1 km$^2$ and it is hard to compare them to areas that are 200 km$^2$ centering the tower which includes many vegetation types. However, we use the vegetation mask to exclude grid cells with different vegetation from the tower's vegetation"*. Seems like it would be more appropriate to keep all of the

pixels within the region because they are not systematically removing airmasses that come from certain wind directions.

**3.6 Discussion of COS and $\Delta^{17}O$**

The timing of this discussion seems weird. The authors mention that these measurements exist but do not actually employ them. As such, I don't feel like this paragraph really adds much value. I'd suggest either removing this paragraph or putting it in the intro.

**3.7 Colorscheme**

As someone who is mildly colorblind, I would prefer the authors use a different color scheme for Figure 1. There's a great discussion on color schemes here: https://personal.sron.nl/~pault/ and examples of how the rainbow color scheme can distort data (see "Good and bad colour schemes compared" near the end). It appears that the authors are using Matlab, I think the "parula" scheme is fairly safe and should be easy to change to. It should just be a matter of adding `colormap(parula)` to the code.

**4  Specific Comments:**

**Page 2, Line 25:** Typo, should be "Congo basin".

**Page 3, Line 10:** Typo? Sentence is confusing: "there is a clear contribution of soil moisture is stress for short vegetation".

---

## Author Comment (AC1) · 4 Nov 2020

GPP seasonal dynamics of Africa

Mengistu et al

Anteneh (antenehgetachew7@gmail.com)

[Figure]

**Authors Response to an interactive comment on "Sun-induced Fluorescence and Near-Infrared Reflectance of vegetation track the seasonal dynamics of gross primary production over Africa"**

Anteneh Getachew Mengistu et al.

November 4, 2020

**Authors response to anonymous Referee #1 Comments:**

We thank the anonymous referee for the time spent to read our manuscript and provide important comments and suggestions. They are enormously constructive and are used to improve the quality of the manuscript. The reply to these comments have lead to new analyses of spatial relations between soil moisture and SIF/NIRv, the addition of a new figure to assess the effect of spatial resolution, and changes to the text to reflect the different response of tropical broadleaf forests to seasonal soil moisture variations. We will respond to all comments in detail and indicate the changes made in the revised manuscript as follows.

**Comment:** The authors compare SIF and NIRv, along with a handful of vegetation
indices, against six flux towers located across the African continent. They use these data to build a linear model of SIF and NIRv to estimate GPP across the continent. I have concerns about the spatial mismatch between eddy covariance measurements and the satellite products used for upscaling. The authors use 0.5 degree satellite imagery and take the further step of aggregating up to 4 degrees (filtering 0.5 degree pixels by dominant land cover type). These average observations are then compared against EC-derived estimates of GPP. Figure 3 suggests that this spatial aggregation significantly influences the temporal correlation between the satellite measurements and GPP estimates. In the case of GH-Ank, the 0.5 degree measurements of NIRv are dramatically different from the 0.05 data (e.g., the 0.05 degree data show much more temporal variability). For ZM-Mon, the shape of the NIRv curve is quite different during the middle of the growing season when comparing 0.5 to 0.05 degree imagery. This is a fairly challenging problem to get around. On the one hand, the authors offer a nice proof of concept that SIF and NIRv can be scaled to GPP using continental scale observations. On the other, higher resolution measurements of SIF are rapidly becoming available (e.g., TROPOMI, as the authors mention) and are already available for NIRv. In fact, a more extensive, global scale analysis of the NIRv-GPP relationship, using tower-scale satellite measurements, has been presented elsewhere (**?**).

**Response:** We share the referee's concern that the spatial mismatch will have some effect on our results and we demonstrated that the use of fine resolution products will improve the relation among these GPP proxies and EC-GPP. Particularly, for sites like GH-Ank and ZM-Mon where the towers vegetation type is different from its surroundings. Despite these limitations, we reverted to the use of GOME-2A SIF for two reasons: 1) Tower GPP data were available for earlier years (for the years before 2014) for most of the towers and therefore the choice of TROPOMI or OCO-2 will not have been good as there were no overlap data between these satellites observation and tower data. 2) Retrieval of SIF from the high spatial resolution of OCO-2 (1.3 x 2 km$^2$) allows a direct comparison with EC measurements. In contrast, due to its smaller swath, OCO-2 has a large repeat period, which restricts its application in
understanding temporal variation in GPP as a monthly mean is restricted to a few data samples. On the other hand, the swath of GOME-2 is so wide (1920 km per orbit) with a coarse spatial resolution (40 x 80 km$^2$) that in principle allows a global coverage of once per 2 days. This allows retrieval of SIF possible at 0.5° grid at monthly resolution with more representative data in each month. We acknowledge that new instruments can in the near future, and partly already now, offer the best of both worlds, and see this as a justification for (rather than a weakness of) the work we present here.

**Comment:** At a minimum, the authors might consider quantifying how the scaling issue affects their modeled estimates of GPP. They could do this by comparing the coefficients of a model derived from 0.05 NIRv data against the 0.5 degree data.

**Response:** Thank you for the suggestion, now we add a comparison of NIRv at finer resolution (0.05 degrees NIRv) with coarse resolution (0.5 degrees NIRv) at the selected flux towers (i.e., GH-Ank, SD-Dem and ZM-Mon) (Fig. 1). The results show a strong correlation and a slope of $\approx$ 1.0 indicating that there is no significant deviation, but a higher slope for ZM-Mon implies the sampling of coarse resolution including responses from different biomes than the tower biome. To show the effect of this scaling on our GPP estimation we add GPP estimation from 0.05 degree NIRv in Fig. 6 of the main text.

[Figure]

```
figure-1.pdf
```

**Fig. 1.** Time series and scatter plot of fine and coarse resolution NIRv.

**Comment:** The authors could also be more descriptive about how they construct their model. How is missing-ness handled? How are clouds screened for? When aggregating to 4 degrees, these details are going to be quite important for understanding how the final satellite signal is constructed.

**Response:** Here we made a direct comparison and thereby we did not apply modifications to these datasets, but we select good quality data from each source (e.g., we selected measured and good quality gap filled data from the EC-tower, and SIF from GOME-2 removes SIF values of high cloud cover and aerosol loading as well as values retrieved for a solar angle greater than 70 degree). Furthermore, we processed the datasets to have the same temporal resolution of XX days/months. Now we add a statement to explain this in the Analysis method section of the main manuscript. "We use good quality data as recommended by each data source, and further we process these datasets to agree in their temporal resolution of **XX** days/months." Change was made on page 6 of line 34.

**Comment:** I appreciated the authors attempt to use their study to draw inferences about the control on productivity at the continental scale. I think this is a type of framing and analysis that has the potential to make the paper a nice contribution to the literature, as opposed to simply demonstrating that the SIF/NIRv-GPP relationship holds regional.

**Response:** We thank the reviewer for his kind appreciation.

**Comment:** Much of the analysis centered on a discussion on the controls of seasonality in photosynthesis. On P15 L2-8 the authors write: "Our analysis showed seasonality of soil moisture strongly controls plant productivity with a weak intervention of available shortwave radiation. . .During saturation, when the soil is very moist, the amount of shortwave radiation significantly impacts productivity, whereas during the growing or

end period of growing seasons vegetation production has a strong proportion to soil moisture." While possibly true, I think this analysis is a little too broad sweeping. Figure 4 shows that broadleaf evergreen forests have a decline in SWR that coincides with declines in precipitation. Personally, I think it would be quite interesting to see if per-pixel anomalies in SIF and/or NIRv track anomalies in SM. I also think that such an analysis would be more informative about mechanism.

**Response:** We thank the reviewer for this suggestion, which we followed up on. We now provide per-pixel temporal correlation of SIF, NIRv, and EVI with soil moisture and precipitation over the vegetated regions of Africa for the years 2007-2016 (see Fig. 2. Lower correlation was observed over the tropical rainforest region, where the monthly average rainfall always exceeds 100mm/month and covered by a broadleaf evergreen forest. This suggests that the seasonal patterns of GPP may have no correspondence with precipitation/soil moisture over this region, which generally has smaller seasonality in GPP and high soil moisture levels compared to non-broadleaf vegetation types. This additional figure was added to our supplementary figures and discussed in the main text.

[Figure]

figure-2.pdf

**Fig. 2.** correlation of SIF a) NIRv c) and EVI e) with root zone soil moisture from GLDAS, and SIF b), NIRv d) and EVI f) with precipitation from GPCC for the years 2007-2016.

**Comment:** Aggregating all the data together across biomes, like in Figure 5, has the potential to hide as much as it reveals, given that averages only reflect the most common SIF-precip/SIF-SWR relationship, as opposed to potentially more complex per-biome or per-pixel relationships.

**Response:** Indeed, the aggregation does not do justice to many of the spatial differences across the landscape, but we feel that a more accurate and per-pixel estimation of SIF/NIRv based GPP needs a more complex process-based modeling. With the aggregation we only aim to show the possibility of inferring aggregated GPP with less computational cost, while possibly still being of use for larger model-intercomparisons such as TRENDY or CMIP6.

**Minor Comments**

**Comment:** P2 L19: "so-called" can have a quite negative connotation. Consider removing.

**Response:** We thank the reviewer for the suggestion. We remove the word "so-called". change is made on page 2 of line 19.

**Comment:** P5 L1-2: "Uncertainties in NIRv are largely due to inaccuracy in measurements of canopy architecture, including the leaf projection function and the clumping index, both strongly vary in time and space (**?**)." I believe that Zeng argues that NIRv carries information about the leaf projection function, as opposed to the leaf projection function causing uncertainty in NIRv measurements.

**Response:** We thank the reviewer for this comment and have removed this sentence.

**Comment:** P7 LUE framework  ď How appropriate is the LUE framework when you normalize by cosine of solar zenith angle. Doesn't that mean the APAR signal

goes away? How should we interpret what is left?

**Response:** We understand the point that the reviewer alludes to, but we believe the equations that we showed are still be valid These LUE frameworks are discussed to motivate that we can create a linear fit between SIF and GPP (**?**). However, the temporal mismatch impacts the correlation between instantaneous SIF and daily GPP. By scaling SIF by the cosine of the solar zenith angle we make sure that we are not so dependent on the position of the sun at the time of observation (a way to adjust the instantaneous SIF observations to a common scale) (**???**)

**Comment:** P13 L3 The manuscript does not address uncertainties in the eddy covariance measurements, so seems unnecessary to spend so much time discussing how the approach is uncertain in tropical context.

**Response:** The statement is to tell the readers that measurement uncertainties are also responsible for the poor correlation with GPP from the GH-Ank tower.

**Comment:** P15 L19: Again, the paper does not use COS, making this discussion feel a little out of place.

**Response:** We move this discussion to the conclusion section to recommend it as another alternative for further study.

**1   References**

Badgley, G., Anderegg, L. D., Berry, J. A., and Field, C. B.: Terrestrial gross primary production: Using NIRV to scale from site to globe, Global change biology, 25, 3731–3740, https://doi.org/10.1111/gcb.14729, 2019.

Hu, J., Liu, L., Guo, J., Du, S., and Liu, X.: Upscaling solar-induced chlorophyll fluorescence from an instantaneous to daily scale gives an improved estimation of the gross primary productivity, Remote Sensing, 10, 1663, https://doi.org/10.3390/rs10101663, 2018.

Joiner, J., Guanter, L., Lindstrot, R., Voigt, M., Vasilkov, A., Middleton, E., Huemmrich, K., Yoshida, Y., and Frankenberg, C.: Global monitoring of terrestrial chlorophyll fluorescence from moderate-spectral-resolution near-infrared satellite measurements: methodology, simulations, and application to GOME-2, Atmospheric Measurement Techniques, 6, 2803–2823, https://doi.org/10.5194/amt-6-2803- 2013, 2013.

Köhler, P., Guanter, L., Kobayashi, H., Walther, S., and Yang, W.: Assessing the potential of sun-induced fluorescence and the canopy scattering coefficient to track large-scale vegetation dynamics in Amazon forests, Remote Sensing of Environment, 204, 769–785, https://doi.org/10.1016/j.rse.2017.09.025, 2018.

Zeng, Y., Badgley, G., Dechant, B., Ryu, Y., Chen, M., and Berry, J. A.: A practical approach for estimating the escape ratio of near-infrared solar-induced chlorophyll fluorescence, Remote Sensing of Environment, 232, 111 209, https://doi.org/10.1016/j.rse.2019.05.028, 2019.

Zhang, Y., Xiao, X., Zhang, Y., Wolf, S., Zhou, S., Joiner, J., Guanter, L., Verma, M., Sun, Y., Yang, X., et al.: On the relationship between sub-daily instantaneous and daily total gross primary production: Implications for interpreting satellite-based SIF retrievals, Remote sensing of environment, 205, 276–289, https://doi.org/10.1016/j.rse.2017.12.009, 2018.

---

## Author Comment (AC2) · 4 Nov 2020

GPP seasonal dynamics of Africa

Mengistu et al

Anteneh (antenehgetachew7@gmail.com)

[Figure]

**Authors Response to an interactive comment on "Sun-induced Fluorescence and Near Infrared Reflectance of vegetation track the seasonal dynamics of gross primary production over Africa"**

Anteneh Getachew Mengistu et al.

November 4, 2020

**Authors response to anonymous Referee #2 Comments:**

We thank the anonymous referee for the time spent to read our manuscript and provide important comments and suggestions. They are enormously constructive and are used to improve the quality of the manuscript. Our reply included two extra analyses and two extra figures that are included and discussed in this document. We will respond to the comments in detail and indicate the changes made in the revised manuscript as follows.

**1 Overview:**

**Overview:** Review of "Sun-induced Fluorescence and Near Infrared Reflectance of vegetation track the seasonal dynamics of gross primary production over Africa" by Mengistu et al. My sincere apologies for the delay in my review.

**Response:** Thank you for the comment, we fully understand the difficulty in thee times to perform tasks like this.

**Overview:** Mengistu et al. present an analysis of SIF, NIRv, EVI, and NDVI over the African continent. They compare these remote-sensing products to flux towers across multiple ecosystems. They find soil moisture to be the dominant driver for much of their data. They also find both SIF and NIRv do a better job of reproducing GPP than EVI or NDVI. Overall, the study is both interesting and useful.

**Response:** We thank the reviewer for kind understanding and acknowledgment that this study is interesting and useful.

**Overview:** My main comments relate to quantitatively assessing the claim that SIF and NIRv are performing better.

**Response:** We demonstrated that SIF and NIRv do a better job than EVI and NDVI in capturing the seasonal dynamics of GPP. However, a performance comparison of SIF and NIRv against VI's (EVI or NDVI) is needed to conclude which one is better than the other. We did not do that in this study, as our focus is mostly on these products given their promise as demonstrated in recent work (**???**).

**Overview:** The figures are reasonably easy to follow and the text is quite well written. I would recommend minor revisions.

**Response:** We thank the reviewer for his kind appreciation. We provide response to his minor revisions.

**2   Major comments:**

Both of my main comments relate to the claim that SIF and NIRv are performing better than NDVI and EVI.

**2.1   Are SIF and NIRv actually performing demonstrably better?**

**Comment:** In the conclusions, the authors state: "The mean climatology of SIF and NIRv agrees widely with the MPI-BGC GPP products in large parts of Africa, confirming their values as a more robust GPP proxy than the commonly used MODIS vegetation indices. However, it's not clear to me that 1) agreement with the MPI-BGC GPP esti- mate is the metric by which to justify that statement and 2) that the differences between some of the products is not within the noise. The latter statement implies to me that SIF and NIRv are both performing comparably well while NDVI and EVI are performing measurably worse.

**Response:** In this particular study, we did not aim to provide a performance compari- son of SIF/NIRv against the NDVI or EVI. Our target is to assess how good are SIF and NIRv to track the seasonal dynamics of GPP over Africa. We agree with the reviewer that MPI-BGC GPP products are not the metric. Moreover, we demonstrated that SIF and NIRv have better correlation to tower GPP than MPI-BGC (See Fig. 2 of the main text, for towers CG-Thc and SN-Dem). Therefore, it is clear that our conclusion need to be rephrased and we now edit the statement and read as "The mean climatology of SIF and NIRv correlates well with GPP from EC-towers, confirming their value as a robust GPP proxy."

**Comment:** From examination of Figure 2, it's not clear that is the case. I would cer- tainly agree that NIRv are and SIF are performing better than EVI at CG-Tch, but NIRv and EVI seem nearly identical at GH-Ank while SIF seems to show no real correspondence with the EC data.

**Response:** Our analysis shows that SIF and NIRv have a better correlation than EVI over SN-Dhr, CG-Tch, and ZA-Kru flux towers. We agree that EVI shows better performance for ZM-Mon flux tower. However, for this particular site, both SIF and NIRv also showed a strong correlation >0.96. This strong correlation for all proxies is expected as the region is dominated by a deciduous broadleaf vegetation which has a clear seasonality of leaves up and down.

**Comment:** I'm wondering if it would be possible to set up some hypothesis test to quantify this. Or maybe it could be done via bootstrapping. For example the authors could randomly sample the different datasets and compare them to the EC data and the MPI-BGC data then report an uncertainty on the R values. Or report the fraction of times the different remote-sensing products performed better than each other.

**Response:** We thank the reviewer for this suggestion and we agree that this could be an interesting approach if one wanted to create a ranking of different VI's and their correspondence to towers. Then again, as agreed with the reviewer an agreement to the MPI-BGC GPP is perhaps not the strongest metric and might not lead to a new conclusion. To show our commitment to the reviewer's remark, we do present here a comparison of NIRv, SIF, NDVI, and EVI with GPP combined from the five EC-tower used in this study. Because none of all proxies showed a correspondence to EC-GPP from GH-Ank site, we exclude values from this site. What we see is that interpreting the R-squares are indeed favorable to SIF and NIRv, but these also show a stronger linear correlation and lower root mean squared deviation than the MODIS Vegetation indices (EVI and NDVI) (See Fig. 1). Moreover, it suggests that the SIF-GPP and NIRv-GPP relationship is possibly less dependent on the vegetation type than the NDVI-GPP and EVI-GPP relations (because we aggregate observations from 5 different towers).

This is an interesting result and we are contemplating a way to look deeper into this,

but given the focus of our current work and the uncertain outcome of the proposed bootstrapping ranking effort, we did not pursue this comparison for now.

[Figure]

**Fig. 1.** The relationship of NIRv, SIF, NDVI, and EVI with tower GPP. all correlations have a p-value $< 10^{-6}$.

[Figure]

**2.2 Are SIF or NIRv capturing those downregulation effects due to water availability?**

**Comment:** As the authors note, the MPI-BGC and other models (e.g., VPRM) include terms for water availability. The authors show that root zone soil moisture is both important for GPP across Africa and strongly coupled to SIF/NIRv. Does this mean that SIF and NIRv are responding to changes in water availability?

**Response:** Yes, our hypothesis is that remotely-sensed SIF and NIRv reflect changes in canopy-scale photosynthesis that are caused by (or at least very strongly influenced by) changes in water availability. The nuance here is for the tropical broadleaf biome, where moisture levels seem to remain sufficiently high to make this relation weak, or absent in the data we used.

**Comment:** If so, are they responding in a manner that better approximates GPP than NDVI or EVI? Related to this, does this mean that models using SIF or NIRv would not need a water availability term? Or do they still require one?

**Response:** The target of our study is to test the usefulness of SIF and NIRv and therefore we did not compare the performance of EVI and NDVI in capturing the water availability. At this stage, we do not suggest that water availability terms need to be included in models for SIF or NIRv, but since these proxies reflect such changes in the vegetation, a model of **GPP** would be required to represent the state of soil moisture well. If not, its GPP is likely to not agree with the remotely-sensed SIF and NIRv that we used.

**Comment:** The authors mention that the rapid decrease at CG-Tch starting in June is due to water availability, I would be interested to see a time-series for individual years that show SIF and NIRv tracking this decrease and some measure of water availability. This could be a supplemental figure, but it would be nice to see that it's

actually occurring on individual years.

**Response:** We perform an extra analysis of the timeseries for the years covering 2007-2016 around CG-Tch tower (see Fig. 2). The correlation of SIF, NIRv, and EVI are 0.77, 0.89, and 0.88 with SM and 0.72, 0.64, and 0.64 with precipitation. The lower correlation between SIF and SM is due to SIF responding earlier than the soil gets too dry. Whereas the correlation of EVI and NIRv with SM show the same pattern, this suggests that they respond when the vegetation looses their green color.

```
figure-2.pdf
```

**Fig. 2.** Time series of SIF, NIRv Precipitation and soil moisture.

**3 Minor comments**

**3.1 Wang et al study?**

The authors may want to discuss the Wang et al. (2020; doi:10.1029/2020JG005732) paper. They provide a nice discussion of what these different remote-sensing measures are telling us.

**Response:** We thank the reviewer for bringing this interesting new paper under our attention. It nicely highlights differences between NDVI, VOD and SIF and how they could be used complementary. We have included a reference to this paper in our discussion.

**3.2 Choice of regions?**

**Comment:** In Section 2.1 the authors mention that they split ecosystems across hemispheres (i.e., NH shrubs and SH shrubs). This reviewer is confused about why this decision was made. The precip and insolation will be out of phase, sure, but wouldn't we expect the response to be similar? Or should we expect a different functional relationship?

**Response:** We agree with the reviewer that the behaviour of the vegetation in response to environmental drivers should be similar, and that they are thus not fundamentally different. However, the distinction between the NH and SH is for a more practical reason: it allows us to plot the seasonal variation of environmental drivers and the vegetation response as a function of time (e.g. for precipitation in Fig. 1d).

**3.3 Use of multiple MODIS products?**

**Comment:** I see the authors used MCD43C4 for NIRv but MOD13C2 for EVI and NDVI (Page 6, Line 5)? I'm curious why they didn't use the same set of reflectances and then compute the various indices in a consistent manner?

**Response:** We thank the reviewer for this comment. The reason for doing this, is that MODIS does not provide NIRv in the MOD13C2 dataset, so we calculated it using the BRDF-corrected surface reflectances from MCD43C4, following the steps outlined in (**?**). We have now included a sentence on page 6.

**3.4 LUE models**

**Comment:** Not all of the photons emitted will escape the canopy. Shouldn't there be some canopy escape term in the SIF relationship? If I recall, the Dechant et al. (2020; doi:10.1016/j.rse.2020.111733) paper argued that this canopy escape term is where much of the information is coming from (that we're learning something about structure.

**Response:** We thank the reviewer for this comment. Indeed the photon escape ratio is an important term for the interpretation of the SIF signal. Our LUE framework demonstrate the existence of a linear relationship between SIF and GPP and not to provide a complete empirical relation of GPP and SIF. We define the $LUE_f$ to represent the product of light use efficiency of SIF and fraction of emitted photons escaping the canopy. Now to be consistent with terminology of other literature we re-write our formulation to include the fraction of emitted photons escaping the canopy separately. "where $LUE_f$ is the effective light use efficiency of SIF and $f_{esc}$ is the fraction of SIF photons escaping the canopy (**??**)."

[Figure]

**3.5   Flux tower comparison**

**Comment:** Presumably the flux towers can be influenced by different types of vegetation within the footprint of the tower, why do the authors remove those pixels from the satellite data: "The flux towers have a footprint of about 1 km$^2$ and it is hard to compare them to areas that are 200 km$^2$ centering the tower which includes many vegetation types. However, we use the vegetation mask to exclude grid cells with different vegetation from the tower's vegetation". Seems like it would be more appropriate to keep all of the pixels within the region because they are not systematically removing airmasses that come from certain wind directions.

**Response:** "We assume that there is one dominant vegetation type within the small footprint of the flux towers ($\sim$ 1-2 km$^2$). The combination of satellite pixels surrounding the tower cover a larger area and can contain different vegetation types. Therefore, to make the comparison between satellite (with possibly multiple vegetation types) and the flux tower (with a single dominant vegetation type) more representative, the filtering of satellite pixels is required."

**3.6   Discussion of COS and $\triangle^{17}$O**

**Comment:** The timing of this discussion seems weird. The authors mention that these measurements exist but do not actually employ them. As such, I don't feel like this paragraph really adds much value. I'd suggest either removing this paragraph or putting it in the intro.

**Response:** We move this discussion to conclusion section to recommend it as another alternative for further study.

**3.7 Colorscheme**

**Comment:** As someone who is mildly colorblind, I would prefer the authors use a different color scheme for Figure 1. There's a great discussion on color schemes here: https: //personal.sron.nl/ pault/ and examples of how the rainbow color scheme can distort data (see "Good and bad colour schemes compared" near the end). It appears that the authors are using Matlab, I think the "parula" scheme is fairly safe and should be easy to change to. It should just be a matter of adding colormap(parula) to the code.

**Response:** We thank the reviewer for the suggestion. We agree and changed the colorscheme as suggested by the referee.

**4 Specific Comments:**

**Comment:** Page 2, Line 25: Typo, should be "Congo basin".

**Response:** Thank you for catching this. Corrected. change was made on page 2 of line 25.

**Comment:** Page 3, Line 10: Typo? Sentence is confusing: "there is a clear contribution of soil moisture is stress for short vegetation".

**Response:** Thank you for indicating this, we rephrased the sentence into: "there is a clear contribution of soil moisture stress for the changing photosynthetic capacity of leaves as a function of age in broadleaf vegetation..." change was made on page 3 of line 10.

**5 References**

Badgley, G., Field, C. B., and Berry, J. A.: Canopy near-infrared reflectance and terrestrial photosynthesis, Science advances, 3, e1602 244,https://doi.org/10.1126/sciadv.1602244, 2017.

Badgley, G., Anderegg, L. D., Berry, J. A., and Field, C. B.: Terrestrial gross primary production: Using NIRV to scale from site to globe, Global change biology, 25, 3731–3740, https://doi.org/10.1111/gcb.14729, 2019.

Damm, A., Guanter, L., Paul-Limoges, E., Van der Tol, C., Hueni, A., Buchmann, N., Eugster, W., Ammann, C., and Schaepman, M. E.: Far- red sun-induced chlorophyll fluorescence shows ecosystem-specific relationships to gross primary production: An assessment based on observational and modeling approaches, Remote Sensing of Environment, 166, 91–105, https://doi.org/10.1016/j.rse.2015.06.004, 2015.

Dechant, B., Ryu, Y., Badgley, G., Zeng, Y., Berry, J. A., Zhang, Y., Goulas, Y., Li, Z., Zhang, Q., Kang, M., et al.: Canopy structure explains the relationship between photosynthesis and sun-induced chlorophyll fluorescence in crops, Remote Sensing of Environment, 241, 111 733, https://doi.org/doi:10.1016/j.rse.2020.111733, 2020.

Doughty, R., Köhler, P., Frankenberg, C., Magney, T. S., Xiao, X., Qin, Y., Wu, X., and Moore, B.: TROPOMI reveals dry-season increase of solar-induced chlorophyll fluorescence in the Amazon forest, Proceedings of the National Academy of Sciences, 116, 22 393–22 398, https://doi.org/10.1073/pnas.1908157116, 2019.

Koren, G., van Schaik, E., Araújo, A. C., Boersma, K. F., Gärtner, A., Killaars, L., Kooreman, M. L., Kruijt, B., van der Laan-Luijkx, I. T.,von Randow, C., et al.: Widespread reduction in sun-induced fluorescence from the Amazon during the 2015/2016 El Niño, Philosophical Transactions of the Royal Society B: Biological Sciences, 373, 20170 408, https://doi.org/10.1098/rstb.2017.0408, 2018.

---

## Author Response (AR3)

5  by the Editorial Board to explain ourselves and to modify the sentence, and to include a citation. This finding agrees with the comment from the reviewer:

**Comment: P16; L11-14 appear to be duplicated, verbatim, from an unreferenced source.**

10  **Our response:** We sincerely apologize for this verbatim duplication and acknowledge that the original source of this phrasing is the thesis of Dr Badgley. There is no excuse for them to appear in our manuscript without quotation or reference. As far as we can back-trace it appears these sentences were once inserted into an early draft version as a reminder/primer of the information we meant to convey. After several months of iterating, they became part of the final text without any of the authors spotting the grave mistake we made. It was an honest mistake, and we had no malicious intent, and further checks of the

15  document revealed that all other text is original.

In a next version, we will remove the sentence and ensure proper citation of the work of Dr Badgley. Being extra careful, we also modify our wordings on Page 4 line 24-26 and Page 6 of line 33 which were not copied but still bare some resemblance to other work.

20  We kindly thank the Editorial Board for their understanding and the chance to rectify our mistake.